# Phytochemistry and Bioactivity of Essential Oil and Methanolic Extracts of *Origanum vulgare* L. from Central Italy

**DOI:** 10.3390/plants14162468

**Published:** 2025-08-09

**Authors:** Francesca Fantasma, Marco Segatto, Mayra Colardo, Francesca Di Matteo, Maria Giovanna Chini, Maria Iorizzi, Gabriella Saviano

**Affiliations:** Department of Bioscience and Territory, University of Molise, C.da Fonte Lappone snc, 86090 Pesche, IS, Italy; marco.segatto@unimol.it (M.S.); mayra.colardo@unimol.it (M.C.); f.dimatteo@studenti.unimol.it (F.D.M.); mariagiovanna.chini@unimol.it (M.G.C.); iorizzi@unimol.it (M.I.)

**Keywords:** *Origanum vulgare* L., chemical characterization, GC-MS, UHPLC, antioxidant activity, polyphenols, cell culture

## Abstract

*Origanum vulgare* L. is an important aromatic plant traditionally used in folk medicine since ancient times. Its growing interest for the scientific community is mainly attributed to its distinctive chemical profile, which includes bioactive compounds, such as polyphenols (phenolic acids and flavonoids) and volatile compounds (essential oil). These components collectively contribute to oregano’s wide spectrum of biological activities. In this study, the volatile components of the essential oil (WEO_OR) and the polyphenolic fraction of the methanolic extract (ME_OR) obtained from leaves and inflorescences of wild *Origanum vulgare* collected in central Italy were characterized using GC-MS and UHPLC-DAD, respectively. Carvacrol was identified as the major compound in the essential oil, while rosmarinic acid was predominant in the methanolic extract. A comparative analysis was also carried out with a commercially available essential oil (CEO_OR), aiming to evaluate potential differences in chemical composition and antioxidant activity (DPPH, ABTS, and FRAP assays). ME_OR showed the strongest antioxidant activity (DPPH IC_50_ = 0.052 mg mL^−1^; ABTS = 3.94 mg TE mL^−1^; FRAP = 30.58 mg TE g^−1^), followed by CEO_OR (DPPH IC_50_ = 0.45 mg mL^−1^; ABTS = 9.57 mg TE mL^−1^; FRAP = 7.33 mg TE g^−1^), while WEO_OR displayed the lowest values (DPPH IC_50_ = 1.54 mg mL^−1^; ABTS = 0.10 mg TE mL^−1^). Furthermore, ME_OR and WEO_OR were tested in vitro using the human hepatoblastoma cell line HepG2 to assess their potential biological activities related to cell survival and oxidative stress. The results indicated that at the tested doses, neither the ME nor the EO showed significant toxicity, as evidenced by the unchanged proliferation rate of HepG2 cells. However, the ME at low doses (50 and 100 μg mL^−1^) and the EO (0.005%), administered as a pre-treatment, exhibited a protective effect against oxidative stress, as inferred from the reduction in 8-OHdG levels, a marker of oxidative damage to nucleic acids.

## 1. Introduction

*Origanum vulgare* L., commonly known as oregano, is an herbaceous plant belonging to the Lamiaceae family, native to the Mediterranean region and western Eurasia [1]. The use of oregano dates back to ancient times; early civilizations employed it both as a culinary herb and as a traditional remedy for various infectious diseases and types of pain [2]. The plant is currently recognized for its diverse applications in medicine, gastronomy, and agriculture. Oregano is especially appreciated for its volatile oil constituents, which significantly contribute to its characteristic aroma and flavor, widely exploited in the food industry [3]. Species of the *Origanum* genus are known for their chemodiversity, characterized by varying combinations of cymyl compounds, sabinyl compounds, and acyclic components, such as linalool and linalyl acetate. Based on these chemotypes, oregano is typically classified into three major groups [4,5].

The essential oil of *O. vulgare* has demonstrated a variety of pharmacological properties, including antibacterial, antiviral, anti-inflammatory, and antioxidant activities [6,7,8]. These bioactivities are mainly attributed to its primary constituents, carvacrol and thymol [9]. Like other aromatic herbs, oregano leaves and inflorescences are also rich in phenolic compounds, including flavonoids and phenolic acids, which are responsible for additional biological effects, such as antioxidant, anti-inflammatory, antidiabetic, antiviral, cytotoxic, and antitumor activities [10,11,12]. Among these, rosmarinic acid and carvacrol have been shown to exert significant antioxidant effects by scavenging reactive oxygen and nitrogen species—mechanisms thought to cause many of the health-promoting properties of the plant [13,14,15].

Several studies have reported the chemical characterization of different species and subspecies of *O. vulgare*, particularly focusing on essential oils from southern regions of the Italian peninsula, such as Calabria [16], Sicily [17], and Campania [18]. However, comprehensive phytochemical characterization of hydroalcoholic extracts from Italian *O. vulgare* species remains limited.

The present study aims to characterize and compare the volatile components of *O. vulgare* essential oils using GC-MS, analyzing two different samples: (i) a wild plant species collected in the Lazio region during its balsamic period and (ii) a commercial edible sample “Officina delle Erbe” purchased from an herbalist shop.

Furthermore, the polyphenolic profile of the methanolic extract (ME_OR) of the wild plant was assessed through UHPLC-DAD. The antioxidant activities of both essential oil samples and the methanolic extract were evaluated using three in vitro assays: DPPH (2,2-diphenyl-1-picrylhydrazyl), ABTS (2,2-azinobis (3-ethylbenzothiazoline-6-sulfonic acid diammonium salt)), and FRAP (Ferric Ion reducing Antioxidant Power).

Finally, we evaluated the possible biological activity of the compounds in the context of cell survival and oxidative stress. As an experimental model, we used HepG2, a robust cell line that ensures experimental reproducibility, as it is commonly employed to study cytotoxicity and response to oxidative stress [19,20,21].

## 2. Results

### 2.1. Essential Oil Yield and Compositions

The leaves and inflorescences of wild *O. vulgare*, collected in central Italy (Lazio region) during the balsamic period, were subjected to steam distillation, yielding an essential oil (WEO_OR) with a 0.3% yield, based on the initial dry weight of 50 g.

This study was complemented by the characterization of a commercial edible of an *O. vulgare* essential oil sample purchased from the herbalist shop “Officina delle Erbe”.

Table 1 presents the chemical composition of both essential oils, including experimental retention indices, which were compared to literature values [22], as well as percentage compositions and abbreviations for the various terpene classes. The compounds are listed according to their elution on a Rtx^®^-5 Restek capillary column. A total of 41 components were identified in the wild *O. vulgare* EO, and 28 components were identified in the commercial sample, corresponding to 96% and 99.41% of the total chromatographic area, respectively.

This analysis revealed significant variability in some of the major constituents between the two samples.

In both EOs, oxygenated monoterpenes represent the most abundant class (38.36% and 81.94%), followed by monoterpenes (32.93% and 14.02%), sesquiterpenes (20.17% and 2.85%), and oxygenated sesquiterpenes (2.71% and 0.19%) (Table 2).

The main oxygenated monoterpenes identified include monocyclic oxygenated monoterpenes (MMOs), with carvacrol being the most abundant component in both oils, albeit at different concentrations: 19.17% in the wild *O. vulgare* essential oil and 75.95% in the commercial sample. Thymol, which is present only in the commercial essential oil, is also included in this group, accounting for 4.39%. Additionally, aliphatic oxygenated monoterpenes (AMOs) are represented, particularly linalool and linalyl acetate. Linalool is more abundant in the wild oregano oil (9.7%) and less so in the commercial oil, while linalyl acetate is present exclusively in the wild oregano oil at a concentration of 4.18%. Among the monoterpenes, the monocyclic monoterpenes are the most prevalent in both EOs, with γ-terpinene (17.25% and 3.70%), *p*-cymene (3.47% and 6.81%), and α-terpinene (3.72% and 1.18%) being the major constituents. Figure 1a,b show the GC-MS of the wild EO and commercial EO, respectively.

The GC-MS chromatograms shown in Figure 1 illustrate the differences in the composition of the two essential oils. The essential oil from central Italy (Figure 1a) was dominated by carvacrol (19.17%), followed by γ-terpinene (17.25%), linalool (9.7%), germacrene A (7.11%), and β-caryophyllene (4.79%). In contrast, the commercial oil (Figure 1b) was predominantly composed of carvacrol (75.95%), followed by *p*-cymene (6.81%), thymol (4.39%), and γ-terpinene (3.70%). These chromatograms clearly illustrate the stark differences in the composition of the two oils, with the commercial oil being much more uniform and primarily composed of carvacrol (Appendix A shows an enlarged version).

### 2.2. UHPLC-DAD Analysis

Ultra-performance liquid chromatography (UHPLC) analysis was performed on the ME_OR extract to determine the content of its most characteristic polyphenols. Figure 2 shows the UHPLC-DAD chromatograms recorded at four different wavelengths: 260, 280, 320, and 360 nm. Partial identification of the components in ME_OR was carried out by comparing their retention times with those of twenty commercial standards.

In total, fifteen compounds were identified and quantified using known standards for calibration (Figure 3), as described in the Materials and Methods, Section 4.6. In Appendix A are reported the HPLC-DAD single chromatograms of 15 phenolic standards.

The quantification results for eight phenolic acids (gallic acid, protocatecuic acid, 4-hydroxybenzoic acid, chlorogenic acid, vanillic acid, caffeic acid, *p*-coumaric acid, and rosmarinic acid), five flavonoids (catechin, rutin, naringin, quercetin, and naringenin), one phenolic aldehyde, vanillin, and one phenolic oxygenated monoterpene, carvacrol, are presented in Table 3. Rosmarinic acid was the most abundant compound (38.8 ± 2.8 mg g^−1^), followed by gallic acid (13.1 ± 1.3 mg g^−1^), vanillin (8.1 ± 0.6 mg g^−1^), rutin (5.2 ± 0.5 mg g^−1^), and naringin (4.4 ± 0.3 mg g^−1^). Several other compounds were detected in smaller amounts, while some were present only in trace amounts. Additionally, a number of unidentified peaks were observed, which may correspond to glycosylated polyphenols commonly found in the species under investigation [10,23,24]. All phenolic compounds showed linearity over the tested concentration ranges, with R^2^ values consistently above 0.9983, confirming the method’s strong analytical performance (Table 3). All calibration curves of the 15 single standards are shown in Appendix A.

### 2.3. Antioxidant Activity and Polyphenolic Content

The antioxidant activity of the three oregano-derived samples (ME_OR (methanolic extract), CEO_OR (commercial essential oil), and WEO_OR (wild oregano essential oil)) was evaluated using DPPH, ABTS, and FRAP assays, with the results presented in Table 4. ME_OR consistently exhibited the strongest antioxidant activity, as evidenced by its significantly lower DPPH IC_50_ value (0.052 mg mL^−1^). This finding is in agreement with the results of Parra et al. [25], who reported similar IC_50_ values for oregano extracts. ME_OR also showed superior ABTS activity (IC_50_ 0.044 mg mL^−1^ and 3.94 mg TE mL^−1^) and higher FRAP capacity (30.58 mg TE g^−1^), underscoring its potential as a potent free radical scavenger and reducing agent.

In comparison, CEO_OR exhibited moderate antioxidant activity, with an ABTS value of 9.57 mg TE mL^−1^. However, its higher DPPH IC_50_ value (0.45 mg mL^−1^) and lower FRAP (7.33 mg TE g^−1^) suggest a less pronounced antioxidant effect. WEO_OR showed the weakest antioxidant performance, with a DPPH IC_50_ of 1.54 mg mL^−1^ and minimal ABTS activity (0.10 mg TE mL^−1^). The absence of FRAP data for WEO_OR limits further comparison; however, these results suggest that wild oregano essential oil has a lower overall antioxidant potential when compared to the other samples.

The observed antioxidant activities are supported by the total phenolic and flavonoid content measured for the methanolic extract (ME_OR), as shown in Table 5. The total polyphenols in ME_OR were determined to be 75.49 ± 0.9 mg GAE g^−1^, which is consistent with the findings of Yan et al. [26], who reported comparable values for oregano extracts.

Additionally, ME_OR contained 147.2 ± 2.1 mg QUE g^−1^ of flavonoids and 34.7 ± 0.5 mg CAE g^−1^ of flavonoids, further supporting its strong antioxidant properties. These values indicate a high concentration of bioactive compounds, particularly phenolic acids and flavonoids, which are well known for their antioxidant activity.

In conclusion, the methanolic extract of oregano (ME_OR) emerged as the most potent antioxidant, likely due to its higher concentration of polyphenols and flavonoids. The commercial oregano essential oil (CEO_OR) exhibited moderate activity, while the wild oregano essential oil (WEO_OR) demonstrated the lowest antioxidant potential. The variations in antioxidant capacity can likely be attributed to differences in the chemical composition and concentration of active compounds, such as polyphenols, flavonoids, and essential oils, in each sample.

In conclusion, the comparative antioxidant analysis highlighted marked differences between the two essential oils. Despite its higher carvacrol content, the commercial oil (CEO_OR) showed only moderate antioxidant activity compared to the methanolic extract (ME_OR), while the wild oil (WEO_OR) displayed the lowest performance in all assays. These differences underline the importance of not only chemical composition but also matrix complexity in determining antioxidant potential.

### 2.4. Biological Activity of the HepG2 Cell Line

The ability of ME_OR to affect HepG2 cell viability was then evaluated. Given the limited data available in the literature on the dosage and the biological activity elicited by ME_OR, a dose–response experiment was conducted using final concentrations of 50, 100, and 150 µg mL^−1^. At 48 h after treatment, no significant changes in cell number were observed at any of the tested doses, indicating that ME_OR does not affect cell proliferation or induce cytotoxicity under these conditions (Figure 4A).

Next, we investigated the putative ability of ME_OR to protect against oxidative-stress-induced cell death. As expected, treatment with hydrogen peroxide (H_2_O_2_) significantly reduced cell viability.

However, ME_OR pre-treatment did not rescue cell viability. Notably, the highest dose of ME_OR (150 µg mL^−1^) further decreased the cell numbers in the presence of H_2_O_2_, suggesting that elevated concentrations may enhance sensitivity to oxidative damage (Figure 4B). To further explore the effects of ME_OR on redox disbalance, we examined levels of 8-hydroxy-2-deoxyguanosine (8-OHdG), a well-established marker of oxidative damage to nucleic acids.

As expected, H_2_O_2_ treatment led to a significant increase in 8-OHdG immunoreactivity, which was markedly attenuated by low and intermediate doses of ME_OR. In contrast, 150 µg mL^−1^ of ME_OR led to even higher 8-OHdG levels than H_2_O_2_ alone, suggesting that higher concentrations may exacerbate oxidative stress (Figure 4C).

Comparable results were obtained when evaluating the biological effects of WEO_OR, tested at a 0.005% concentration, as already reported in previous findings [27]. WEO_OR did not affect HepG2 cell viability under either normal (Figure 5A) or oxidative stress conditions (Figure 5B). However, WEO_OR significantly mitigated the H_2_O_2_-induced increase in 8-OHdG levels (Figure 5C), demonstrating a robust antioxidant effect.

## 3. Discussion

Characterization by GC-MS of the essential oil from oregano grown in central Italy (Lazio region) and the “commercial oil” of oregano confirmed that carvacrol, a phenolic monoterpene, was the most abundant component, albeit at different concentrations. Based on the chemical composition, it can be established that both essential oils belong to the carvacrol chemotype.

In the commercial essential oil, thymol was present in addition to carvacrol, along with their biogenetic precursors, such as **γ**-terpinene and *p*-cymene. These compounds are associated with the “cimyl” pathway, typically found in *O. vulgare* subspecies rich in essential oils, such as *O. vulgare* subsp. *hirtum* (also known as Greek oregano). Furthermore, the presence of both thymol and carvacrol suggests that the biosynthesis of these two phenolic monoterpenes may be related. This supports Novak’s hypothesis that thymol and carvacrol are produced from **γ**-terpinene and *p*-cymene (biogenetic precursors) via two independent hydroxylases [28].

The wild oregano essential oil from the Lazio region highlighted carvacrol (19.2%) as the main component, followed by a significant amount of **γ**-terpinene (17.3%), compounds associated with the cymyl pathway. Additionally, linalool (9.7%), linalyl acetate (4.17%), associated with the acyl pathway, germacrene A (7.11%), and β-cariofillene (4.79%) were also present. These latter two sesquiterpenes were found in relatively higher concentrations compared to other sesquiterpenes characterized in the same oil. Characterization of essential oils from different regions of Italy has shown that phenolic monoterpenes constitute the most abundant components.

Specifically, the characterization of essential oils from southern Sicily revealed that *O. vulgare* L. subsp. *viridulum* (or *O. heracleouticum*) had a high percentage of thymol, followed by its biogenetic precursors, while the essential oil obtained from the hybridization of *O. vulgare* subsp. *viridulum* x *O. vulgare* subsp. *hirtum* had a high percentage of carvacrol [17,18].

In the Calabria region (Italy), on the other hand, essential oils from different oregano populations showed that carvacrol prevailed in some cases and thymol in others [16]. When comparing the chemical composition of the Lazio region essential oil with those mentioned above, all shared the predominance of a phenolic monoterpene. However, the Lazio essential oil was distinguished by its relatively high percentages of linalool, germacrene A, and β-caryophyllene, which were either present in trace amounts or completely absent in the essential oils from the other regions. Differences in the chemical composition and yield of essential oils, even within the same species, can be attributed to various factors, including climate, altitude, geographical location, harvest time, and growth stage [29].

The plant matrix used for the extraction of the Lazio essential oil was also used to obtain a methanolic extract, which underwent antioxidant activity assays. IC_50_ values obtained through the DPPH assay showed that the methanolic extract exhibited higher antioxidant activity (52.0 μg mL^−1^) than the commercial essential oil (IC_50_ 450 μg mL^−1^). When compared with literature values, the methanolic extract showed lower antioxidant activity than the methanolic extract of *Origanum vulgare* from Turkey (IC_50_ 9.9 μg mL^−1^) [30] and the aqueous extract of Canadian oregano (IC_50_: 26.7 μg mL^−1^) [31] but greater activity than the methanolic extract of oregano from southeastern Romania (IC_50_ 83.95 μg mL^−1^) [32]. The IC_50_ obtained for the commercially sourced essential oil (IC_50_ 450 μg mL^−1^) was similar to that obtained for the essential oil of oregano from Serbia (IC_50_ 590 μg mL^−1^) [33] and the value obtained for a commercial essential oil from Canada (IC^50^ ~500 μg mL^−1^) [30]. Instead, the value obtained for the essential oil extracted from wild oregano, 1.54 mg mL^−1^, partially matched literature data. Interestingly, the commercial oil, with 76% carvacrol, demonstrated a strong ability to inhibit the ABTS radical cation, in contrast to the wild oil with only 19% carvacrol. Moreover, the essential oil COE_OR showed higher antioxidant activity than the methanolic extract in the ABTS assay, possibly due to stronger interactions between oxygenated monoterpenes and the ABTS radical cation [34,35,36]. The FRAP assay, in contrast, showed that the methanolic extract had a good ferric ion reduction ability, in line with the DPPH assay and its polyphenolic content. The strong scavenging action of the methanolic extract was attributed to its content of polyphenolic compounds.

Total polyphenol content (TPC) was 75.49 ± 0.9 mg GAE g^−1^, which was higher than that of oregano methanolic extract from Romania (TPC: 67.8 ± 3.41 mg GAE g^−1^) [37] but lower than an ethanolic extract of *O. vulgare* from the same country (TPC: 94.69 ± 4.03 mg GAE g^−1^) [38]. The total flavonoid content (TFC), expressed as catechin equivalents, was 34.7 ± 0.5 mg CAE g^−1^, significantly higher than that found in the ethanolic extract from Armenia (TFC: 3.9 ± 0.7 mg CAE g^−1^) [39].

The scavenging action of the hydroalcoholic extract is usually due to the synergistic action among compounds that fall into the polyphenol group, particularly phenolic acids and flavonoids. According to the literature, hydroalcoholic oregano extracts contain significant amounts of phenolic acids, such as rosmarinic acid, caffeic acid, vanillic acid, and chlorogenic acid, as well as flavonoids like naringenin, luteolin, and apigenin, often in their glycosidic forms [25]. These phenolic profiles can vary depending on geographical and environmental conditions, making them useful for distinguishing among oregano chemotypes. Even within the same species, genotypes grown in different locations can produce significantly different profiles [40]. For example, both the ethanolic extracts of Greek oregano and common oregano contain phenolic acids (e.g., rosmarinic, caffeic, protocatechuic) and flavonoids (e.g., naringenin, luteolin 7-O-glucoside, apigenin 7-O-glucoside), with differences mainly in relative concentration, though both are rich in rosmarinic acid [41].

In this study, UHPLC-DAD analysis of the methanolic extract (ME_OR) revealed that rosmarinic acid was the most abundant phenolic compound (38.8 ± 2.8 mg g^−1^ DW), followed by gallic acid (13.1 ± 1.3 mg g^−1^ DW), vanillin (8.1 ± 0.6 mg g^−1^ DW), rutin (5.2 ± 0.5 mg g^−1^ DW), and naringin (4.4 ± 0.3 mg g^−1^ DW). Minor components included protocatechuic acid, caffeic acid, vanillic acid, 4-hydroxybenzoic acid, *p*-coumaric acid, and chlorogenic acid, along with catechin, quercetin, naringenin, and carvacrol.

Overall, the comparison between WEO_OR and CEO_OR revealed significant differences both in composition and functional properties. The commercial oil, characterized by a simpler profile with higher carvacrol concentration, showed better ABTS scavenging activity, while the wild oil, richer in terpene diversity and sesquiterpenes, showed limited antioxidant capacity, but it might exhibit other biological effects. These differences may reflect the natural variability of wild oregano and the standardized nature of commercial formulations.

The chemical composition of essential oils can vary significantly depending on the plant’s chemotype. In our study, WEO_OR exhibited a carvacrol chemotype as the dominant compound alongside a broader spectrum of sesquiterpenes and minor constituents, whereas the commercial oil showed a mixed profile dominated by carvacrol.

Concerning the most abundant compound, several findings have demonstrated that rosmarinic acid efficiently counteracts oxidative stress in diverse pre-clinical experimental models. Interestingly, it has been reported that rosmarinic acid exerts its antioxidant effects primarily through the activation of the nuclear factor erythroid 2-related factor 2 (NRF2) [42], one of the master transcription factors governing redox homeostasis [43]. Although the upstream mechanisms are still not completely elucidated and may vary based on the specific physiopathological context, it has been extensively demonstrated that rosmarinic acid can promote NRF2 stabilization/activation, which migrates to the nucleus to promote the expression of antioxidant response element (ARE)-driven genes, such as heme oxygenase 1 (HO-1) and Glutamate-Cysteine Ligase Modifier Subunit (GCLM), thereby enhancing intracellular defense mechanisms [44,45]. Accordingly, rosmarinic acid administration restores GSH/GSSG balance and strongly suppresses ROS production [46]. Collectively, this evidence highlights that rosmarinic acid contrasts oxidative stress not only by acting as a direct scavenger but also by upregulating the endogenous antioxidant systems.

The methanolic extract and essential oil of Lazio were also tested on HepG2 cell culture to evaluate biological effects related to cell viability and oxidative stress. The results showed that at the doses used, neither the methanolic extract nor the essential oil, per se, showed obvious toxicity, as suggested by the unchanged proliferation rate of HepG2 cells. On the other hand, it appeared from the data reported in the literature that both the hydroalcoholic extract and the essential oil of oregano are able to manifest a significant anti-proliferative and cytotoxic action in various tumor cell lines.

In particular, the methanolic extract showed an antiproliferative and cytotoxic effect in colon adenocarcinoma (Caco-2) cells treated with increasing concentrations in the range of 100 to 500 μg mL^−1^. While low doses reported no significant effects, gradually increasing the dose resulted in a reduction in cell viability [47].

Similarly, *Origanum vulgare* essential oil was able to reduce HepG2 viability in a dose-dependent manner compared to controls in a concentration range of 25–800 μg mL^−1^. Another report supports the cytotoxic activity of the essential oil on HepG2 [48]. It has also been shown that *Origanum onites* essential oil decreases cell viability from doses of around 0.008%. The IC_50_ value was determined to be around 0.009%, and cell growth was inhibited in a dose- and time-dependent manner [49]. The discrepancy between the results obtained in this study and those obtained in previous work may be attributed to differences in the analytical method employed. Indeed, in the present work, cell viability was directly assessed by cell counting, whereas earlier studies assessed cell viability through MTT assay, which estimates cell viability based on mitochondrial activity. However, reduced mitochondrial activity does not always correspond to a decrease in cell viability. Indeed, it is well-established that certain compounds can trigger important metabolic rearrangement whereby glycolysis is primarily used as an energy source rather than cellular respiration. This metabolic shift from oxidative phosphorylation to glycolysis is a common feature of cancer cells and frequently observed in hepatic cells under both physiological and pathological contexts [50,51,52].

In our study, neither methanolic extract nor essential oil, at the doses used, increased cell viability upon exposure to pro-oxidant conditions. Interestingly, the higher dose of the methanolic extract contributed to enhanced cell death through mechanisms that are still not explored. In contrast, the methanolic extract at low doses (50 and 100 μg mL^−1^), as well as the essential oil, showed a protective effect against oxidative stress, as highlighted by the reduction in levels of 8-OHdG, a marker of oxidative damage to nucleic acids. The molecular mechanisms capable of inducing an antioxidant response have not been identified in this work, but literature data report that the administration of oregano essential oil is capable of exerting protective effects against oxidative stress not only by exerting direct scavenger activity but also by increasing the expression levels of antioxidant enzymes [53].

Furthermore, it must be emphasized that the effects observed in this work on methanolic extract are strictly dependent on the dose. Indeed, at high concentrations, molecules known for their antioxidant action lose their effects and manifest a pro-oxidant action, causing membrane and DNA damage. Conversely, at low concentrations, these molecules can manifest a protective effect on the membrane and DNA, thus acting as antioxidants [54]. In this context, the low-dose methanolic extract (50–100 μg mL^−1^) showed an antioxidant effect, while the higher dose (150 μg mL^−1^) induced an opposite effect, further increasing levels of the oxidative damage marker 8-OHdG, thus acting as a pro-oxidant. Furthermore, the highest concentration tested not only exacerbated the intracellular oxidative stress but also increased the susceptibility of HepG2 cells to H_2_O_2_-induced cytotoxicity. Similar results were also obtained by other research groups that focused their attention on the effects of carvacrol and thymol, two compounds particularly enriched in oregano essential oil. Specifically, an increase in ROS was observed in Caco2 cells after 24 h of exposure to a concentration of 460 μM carvacrol and after 48 h of treatment at 230 μM.

A similar observation was obtained through the administration of a carvacrol–thymol mixture in a 10:1 ratio [55]. Furthermore, other studies have reported that carvacrol at concentrations above 380 μM induces increased levels of the oxidative damage markers MDA and 8-OHdG in lung cancer cells (H1299) 24 and 48 h after administration [56].

## 4. Materials and Methods

### 4.1. Reagents and Chemicals

Acetonitrile, methanol, and water (all HPLC-grade) were purchased from Romil-UpS™ (Romil Ltd., Cambridge, UK). Standards of apigenin (API), kaempferol (KAM), chlorogenic acid (CLA), rosmarinic acid (RA), rutin (RUT), (−)-epicatechin (EPI), (+)-catechin (CAT), carvacrol (CAR), thymol (THY), quercetin (QUE), p-coumaric acid (PCA), naringin (NAR), vanillin (VAN), caffeic acid (CA), hydrated rutin (HRUT), proto-catechuic acid (PRCA), gallic acid (GA), 4-hydroxybenzoic acid (4-HBA), (±)-naringenin (NAN), and vanillic acid (VA) were obtained from CPA Chem (Stara Zagora, Bulgaria) and Dr. Ehrenstorfer Gmbh (Augsburg, Germany). All other reagents and chemicals were of analytical reagent-grade. The EO was characterized by GC-MS using a range of standards: α-pinene, β-pinene, γ-terpinene, linalool, terpinen-4-ol, thymol, and carvacrol (Sigma-Aldrich, St. Louis, MO, USA).

### 4.2. Plant Material

The plant matrix used for the following study consisted of leaves and inflorescences of wild oregano (*Origanum vulgare*) from the Lazio Region in central Italy (41°36′3.132″ N, 12°32′47.291″ E). It was sampled in its balsamic period (June 2022), where the yield of active ingredients is at its maximum.

The plants were authenticated as *Origanum vulgare* by a botanic group, and voucher specimens are kept at the Herbarium of the Department of Bioscience and Territory, University of Molise. The plants were air dried at room temperature and dried in an oven to constant weight (40 °C for 72 h). The sample was divided into two aliquots to undergo two different extractions: steam distillation to obtain *O. vulgare* EO and hydroalcoholic extraction.

### 4.3. Essential Oil Isolation

A defined amount of wild oregano plant material (50 g of leaves and inflorescences) was subjected to steam distillation for 3 h according to the standard procedure outlined in the *European Pharmacopoeia* [57]. Anhydrous sodium sulfate was used to remove traces of water. The distillate (WEO_OR) was stored in a dark vial at 4 °C until analysis.

The commercial oregano essential oil (CEO_OR) was purchased from a certified supplier “Officina delle erbe” and analyzed in its original form after appropriate dilution. No further processing was applied prior to chemical or antioxidant analysis.

### 4.4. GC-MS Analysis, Identification of EO Components

The chemical composition of both wild oregano essential oil (EO) and commercial EO was determined via gas chromatography–mass spectrometry (GC–MS) analysis. The analysis was conducted using a Trace GC Ultra gas chromatography system (Thermo Fisher Scientific, Waltham, MA, USA) equipped with an Rtx^®^-5 Restek capillary column (30 m × 0.25 mm i.d., 0.25 µm film thickness) and coupled with a Polaris Q ion-trap (IT) mass spectrometry (MS) detector (Thermo Fisher Scientific, Waltham, MA, USA). A programmed temperature vaporizer (PTV) injector and a PC equipped with the chromatography software Xcalibur (Thermo Fisher Scientific, Waltham, MA, USA) were used.

The ionization voltage was set to 70 eV and the source temperature to 250 °C. Full scan acquisition in positive chemical ionization mode was conducted over an *m*/*z* range of 40 to 400 a.m.u. at a scan rate of 0.43 scans per second. The column temperature was initially held at 40 °C for 5 min and then increased to 250 °C at a rate of 3 °C/min, followed by an isothermal hold for 10 min. Helium was used as the carrier gas at a flow rate of 1.0 mL min^−1^. A 1 µL aliquot of each sample was dissolved in n-hexane (1:500 n-hexane solution) and injected. The experiment was repeated in triplicate.

The identification of the components in both EOs was based on the comparison of their Kovats retention indices (Exp RI), calculated from the retention times (tR) of a homologous series of n-alkanes (C_8_–C_20_) injected under the same conditions [58,59].

Mass spectrometry (MS) fragmentation patterns of the individual compounds were compared with those in the NIST 2.6 library, as well as the Adams and Wiley 275 mass spectral libraries [60,61].

The relative abundances (%) of the sample components were calculated as the average of the GC peak areas obtained from the triplicate runs, without further correction [62].

### 4.5. Hydroalcoholic Extraction

A hydroalcoholic extract of the plant material was prepared following the procedure described in [14]. Specifically, 0.45 g of dried powder of oregano (leaves and inflorescences) was mixed with 9.0 mL of a methanol/water solution (80:20, *v*/*v*). The mixture was stirred continuously for 45 min at room temperature, followed by 15 min of incubation in the dark. Subsequently, the sample was centrifuged at 5000 rpm for 15 min to collect the supernatant, which was then stored in the dark at 4 °C until further analysis.

### 4.6. UHPLC-DAD Analysis

UHPLC analysis was conducted using a Vanquish UHPLC system (Thermo Fisher Scientific, Waltham, MA, USA), equipped with a binary pump, autosampler, column compartment, and photodiode array detector (DAD).

Chromatographic separation was performed on an Accucore C18 column (150 × 2.1 mm; Thermo Fisher Scientific, Waltham, MA, USA) equipped with an Accucore C18 (10 × 2.1 mm, 2.6 um defender guards pk4 (Thermoscientific, USA)) maintained at 30 °C in a thermostatically controlled column oven.

A gradient elution was carried out at a flow rate of 0.4 mL min^−1^ using 0.1% formic acid in water (solvent A) and acetonitrile (solvent B). The gradient profile is detailed in Table 6. After the run, the system was flushed with 5% solvent B for 4 min to clean the column and then re-equilibrated to initial conditions.

Stock standard solutions of each phenolic compound (1 mg mL^−1^) (see Section 4.1) were prepared by dissolving accurately weighed amounts in UHPLC-grade methanol, with the aid of brief sonication. The temperature of the water bath during sonication was carefully monitored to remain below 30 °C. Stock solutions were stored in the dark at −20 °C until analysis.

Working standard solutions, containing all phenolic compounds, were obtained by diluting the stock solutions with the initial mobile phase to cover specific concentration ranges: 0.2–45 μg mL^−1^. To ensure complete solubilization, the mixtures were vortexed or sonicated immediately after preparation and again prior to injection.

Spectral data were acquired using the DAD at the following wavelengths: 260 nm (PRCA, 4-HBA, VA, RUT), 280 nm (GA, CAT, NAR, NAN, CAR), 320 nm (CLA, CA, PCA, RA), and 360 nm (QUE, VAN). Chromatograms were processed using the Thermo Scientific Dionex Chromeleon 7 Chromatography Data System (version 7.3).

Phenolic compounds were identified by comparing the retention times of sample peaks with those of authentic standards. Calibration curves were generated using five concentration levels, each analyzed in triplicate, and the data were processed using linear regression analysis. For each analyte, the regression equation, slope, intercept, and coefficient of determination (R^2^) were calculated to assess the linearity of the method.

In particular, calibration curves were constructed using four concentration levels within the following ranges, including 0.2–20 μg mL^−1^ for 4-HBA, CLA, PCA, and CAR, 0.5–20 μg mL^−1^ for PRCA, CAT, VA, CA, VAN, RUT, NAR, QUE, and NAN, 5–30 μg mL^−1^ for GA, and 10–45 μg mL^−1^ for RA.

Linearity and the corresponding coefficients of determination (R^2^) were evaluated for each of the 15 phenolic compounds detected (Table 3 and Appendix A). All compounds exhibited excellent linearity, with R^2^ values exceeding 0.9983, indicating a strong correlation between concentration and response within the tested ranges.

Calibration curves were derived using linear least squares regression of the peak areas against the respective concentrations. Linearity was assessed individually for each compound based on the corresponding R^2^ value. The limit of detection (LOD), in this context, was evaluated through visual assessment, as reported in “International Conference on Harmonization, 2023. ICH Q2(R2) Guideline on validation of analytical procedures”, Sections 3.2.3 and 3.2.3.1.

### 4.7. Antioxidant Activity

The antioxidant activity was evaluated for the wild essential oil (WEO_OR), the commercial essential oil (CEO_OR), and the methanolic extract of *Origanum vulgare*.

The essential oils were appropriately diluted prior to analysis. The methanolic extract was prepared by extracting 0.45 g of dried and ground *O. vulgare* plant material in 9 mL of 80% methanol [63].

The mixture was subjected to ultrasonic treatment for 45 min, followed by 15 min of incubation in the dark. The resulting extract was then centrifuged at 5000 rpm for 15 min. The supernatant was collected and stored at −20 °C until analysis. All extractions were performed in triplicate. The antioxidant capacity of the samples was assessed using three complementary methods: DPPH• (1,1-diphenyl-2-picrylhydrazyl) radical scavenging assay, ABTS (2,2′-azino-bis (3-ethylbenzothiazoline-6-sulfonic acid)) radical cation decolorization assay, and FRAP (ferric reducing antioxidant power) assay.

#### 4.7.1. DPPH Radical Scavenging Activity

The DPPH radical scavenging activity of the essential oils was assessed following the method described by Wei et al. [64], with slight modifications. Specifically, the essential oils were diluted in absolute ethanol prior to testing. A volume of 1 mL of the appropriately diluted essential oil was mixed with 1 mL of a freshly prepared ethanolic DPPH• solution (27 μg mL^−1^). The reaction mixture was incubated in the dark for 30 min. DPPH• radicals are reduced upon reaction with antioxidant compounds, resulting in a noticeable color change. This change was quantified by measuring the absorbance at 517 nm using a Shimadzu UV-1601 spectrophotometer (Kyoto, Japan), with ethanol as the blank.

A control was prepared by mixing 1 mL of the DPPH• solution with 1 mL of ethanol.

Ascorbic acid (2.0–1000 μg mL^−1^), known for its antioxidant properties, was used as a positive control and tested under the same conditions.

The antioxidant activity was expressed as IC_50_, defined as the concentration of the sample (mg mL^−1^) required to reduce 50% of the DPPH• radicals. IC_50_ values were calculated through linear regression analysis of the dose–response curves, obtained by plotting the radical scavenging activity against sample concentrations ranging from 0.1 to 1.2 mg mL^−1^.

The radical scavenging activity (%) was calculated using the following equation:Radical Scavenging Activity (%) = [1 − (A_sample_/A_control_)] × 100
where *A_control_* was the absorbance of the DPPH• solution mixed with ethanol and *A_sample_* was the absorbance of the test sample.

All measurements were performed in triplicate, and results are reported as mean IC_50_ values ± standard error (SE).

#### 4.7.2. ABTS Radical Scavenging Activity

With certain modifications, the antioxidant capacity of oregano methanolic extracts was determined following the method described by Re et al. [65]. An ABTS solution (7 mM) and potassium persulfate (K_2_S_2_O_8_, 2.45 mM) were mixed in equal volumes (1:1 *v*/*v*) and allowed to react in the dark at room temperature for 16 h to generate the ABTS radical cation (ABTS•^+^). The resulting radical solution was then diluted with 80% (*v*/*v*) methanol to reach an absorbance of 0.70 ± 0.1 at 734 nm.

For the assay, 150 μL of the extract was mixed with 1350 μL of the ABTS•^+^ solution and incubated in the dark for 5 min. Absorbance was measured at 734 nm using a Shimadzu UV-1601 spectrophotometer.

The ABTS radical scavenging activity of the extract was calculated as the percentage of inhibition using the following equation:% ABTS Inhibition = [1 − (A_sample_/A_control_)] × 100
where *A_control_* is the absorbance of the 150 μL of methanol with 1350 μL of ABTS.

The absorbance values of the samples were compared to a Trolox standard curve, and the antioxidant capacity was expressed as mg Trolox equivalents per mL of sample (mg TE mL^−1^).

#### 4.7.3. Ferric Reducing Power (FRAP)

The ferric reducing antioxidant power (FRAP) of the essential oils was determined following the method described by Benzie et al. [66], with slight modifications.

The FRAP reagent was freshly prepared by mixing 300 mM acetate buffer (pH 3.6), 10 mM TPTZ (2,4,6-tripyridyl-s-triazine) in 40 mM HCl, and 20 mM FeCl_3_·6H_2_O in a 10:1:1 ratio. For the assay, 200 μL of the appropriately prepared extract was added to 1.8 mL of the FRAP reagent. The reaction mixture was incubated at 37 °C for 30 min.

After incubation, absorbance was measured at 593 nm using a UV-1601 spectrophotometer (Shimadzu, Kyoto, Japan). An increase in absorbance at 593 nm indicates greater ferric reducing (antioxidant) capacity of the sample.

A calibration curve was prepared using Trolox standard solutions at increasing concentrations expressed in micromoles per liter (4–25 µM). Antioxidant activity results were expressed as Trolox equivalents in milligrams per gram of sample (mg g^−1^). The conversion from µM to mg g^−1^ was performed based on the molar mass of Trolox (250.29 g mol^−1^), the volume of extract used, and the weight of the sample. Final values were calculated by interpolating sample absorbance values on the standard curve and normalizing the results to the amount of sample extracted.

### 4.8. Polyphenolic Content

The total phenolic content (TPC) of the ME_OR was determined using the Folin–Ciocalteu method, with minor modifications [67].

A volume of 1.0 mL of extract was mixed with 0.5 mL of Folin–Ciocalteu reagent (diluted 1:10). After a reaction time of 10 min, 3.0 mL of 7.5% sodium carbonate solution (Na_2_CO_3_) was added, and distilled water was used to bring the final volume to 5.0 mL.

The mixtures were incubated in the dark at room temperature for 60 min. Following incubation, the solutions were centrifuged at 4000 rpm for 3 min. The supernatants were collected and transferred into cuvettes for absorbance measurement at 765 nm using a Shimadzu UV-1601 VIS spectrophotometer (Shimadzu, Kyoto, Japan).

Phenolic content was quantified using a gallic acid calibration curve (1.0–10 μg mL^−1^), and results were expressed as milligrams of gallic acid equivalents (mg GAE) per gram of sample. All determinations were carried out in triplicate, and values are reported as means ± standard error.

### 4.9. Flavonoid Content

The method described by Kim et al. [67] was used to determine the total flavonoid content of the extracts. Briefly, 80 μL of a 5% NaNO_2_ solution was added to 100 μL of the extract, which had previously been mixed with 1.5 mL of distilled water. After 6 min, 150 μL of a 10% AlCl_3_·6H_2_O solution was added, followed by 500 μL of 1.0 M NaOH solution, and the mixture was left to stand in the dark for another 5 min. Distilled water was then added to bring the final volume to 6.5 mL, and the solution was thoroughly mixed. Absorbance was promptly measured at 510 nm using a UV-1601 spectrophotometer (Shimadzu, Kyoto, Japan) against a blank prepared with the same reagents but without the extract. The results were calculated and expressed as mg of quercetin and catechin equivalents (mg QUE g^−1^ DM and mg CE g^−1^ DM, respectively) using the corresponding calibration curves. All experiments were performed in triplicate, and the results are presented as mean values.

### 4.10. Cell Cultures

HepG2 cells were cultured at 5% CO_2_ in a DMEM medium at high glucose (D6429, Merck Life Science, Milan, Italy), containing 10% (*v*/*v*) fetal bovine serum (FBS, F7524, Merck Life Science, Milan, Italy) and 1% penicillin/streptomycin solution (P06-07100, PAN Biotech, Aidenbach, Germany). Six hours after seeding, HepG2 cells were treated with 50, 100, and 150 µg mL^−1^ of ME_OR or with WEO_OR at a concentration of 0.005%.

To facilitate solubilization in the growth medium, the required amount of oil was first conveyed into FBS (at a final concentration of 10% in DMEM) and then into DMEM. Control cells received DMSO (dilution 1:1000 in cell culture medium) as the vehicle. To induce oxidative stress, cells were exposed to H_2_O_2_ at a dose of 500 µM after a 24 h pre-treatment with either the methanolic extract or the essential oil. Additionally, wells treated with H_2_O_2_ (500 µM) were included as positive controls for oxidative damage.

#### 4.10.1. Cell Viability and Cytoprotection Assessment

The growth curve experiments were assessed by seeding HepG2 cells into 12-well plates (50,000 cells for each well). Six hours after seeding, cells were treated with the different doses of ME_OR or with WEO_OR, and cell counts were performed after 48 h. Cell number was determined manually using a Blutzählkammer THOMA chamber (Merck Life Science, Milan, Italy).

To evaluate the protective effect of the ME_OR and WEO_OR against H_2_O_2_ treatment, the number of nuclei per field was quantified using DAPI staining. Cells were seeded on coverslips and treated as previously described. Subsequently, cells were fixed with 4% paraformaldehyde for 10 min and permeabilized with 0.1% Triton X-100 in PBS for 5 min at room temperature. After DAPI staining (D9542, Merck Life Science, Milan, Italy), the coverslips were mounted with Fluoroshield mounting medium (F6182, Merck Life Science, Milan, Italy) and examined under a confocal microscope (TCS SP8; Leica, Wetzlar, Germany). Images were captured using Leica TCS SP8 equipped with 20× magnification and Leica LAS X software (version 3.5.5) (Leica Camera, Wetzlar, Germany) for Windows 10.

#### 4.10.2. Immunofluorescence and Confocal Analysis

HepG2 cells were seeded at a density of 200,000 cells per well on coverslips and grown in DMEM high glucose with 10% FBS and treated as described above. Thereafter, cells were fixed with 4% paraformaldehyde for 10 min and then permeabilized with 0.1% Triton X-100 in PBS for 5 min at room temperature. Blocking was performed using 3% Bovine Serum Albumin (BSA) in 0.1% Triton X-100/PBS for 1 h. DNA was denatured through incubation with 2 M HCl for 20 min at room temperature. The 8-OHdG (Santa Cruz Biotechnology, Dallas, TX, USA, sc-66036; dilution 1:100) primary antibody was incubated overnight at 4 °C and visualized using Alexa 555 Fluor secondary antibodies (Thermo Fisher Scientific, Waltham, MA, USA). After nuclear staining with DAPI (D9542, Merck Life Science, Milan, Italy), the coverslips were mounted with Fluoroshield mounting medium (F6182, Merck Life Science, Milan, Italy) and visualized under a confocal microscope. Images were captured using Leica TCS SP8 equipped with 20× or 40× magnification and analyzed with Leica LAS X software (version 3.5.5) for Windows 10.

### 4.11. Statistical Analysis

Results presented in this study were expressed as mean ± SD (standard deviation). One-way analysis of variance (ANOVA) or two-way ANOVA was carried out, followed by Tukey’s or Bonferroni’s post hoc test, respectively. *p* < 0.05 was considered statistically significant. Statistical analysis and graph editing were conducted using GraphPad Prism 8.4.2 (GraphPad, La Jolla, CA, USA) for Windows 11.

## 5. Conclusions

The analysis of the terpenes and “terpenoids” in both wild-type and commercial *O. vulgare* oils reveals distinct compositional differences that may influence their biological properties. The wild-type essential oil displayed a greater diversity of monoterpenes, oxygenated monoterpenes, and sesquiterpenes, suggesting a more complex pharmacological profile. These differences could be attributed to factors like geographical origin, harvesting conditions, and species variations. In contrast, the commercial oil showed a higher concentration of carvacrol, which might indicate a more uniform composition, possibly resulting from standardization processes used in commercial production. The higher percentage of oxygenated sesquiterpenes in the wild oil further suggests that it may have broader potential for therapeutic applications, particularly in anti-inflammatory and antioxidant activities. These findings underscore the importance of considering the source and type of essential oil when evaluating its potential health benefits. The wild *O. vulgare* essential oil, with its diverse composition, may offer a wider range of biological activities compared to the more uniform commercial oils. In conclusion, although further studies are needed to fully understand the molecular mechanisms, this work demonstrates that extracts derived from *Origanum vulgare* exhibit significant biological activity, particularly in the context of oxidative stress. It is important to note that dosage plays a critical role in determining the functional properties of these compounds, which can act as antioxidants or pro-oxidants depending on the amount administered. Therefore, future research should include dose–response studies to better assess the biological properties of these compounds.

## Figures and Tables

**Figure 1 plants-14-02468-f001:**
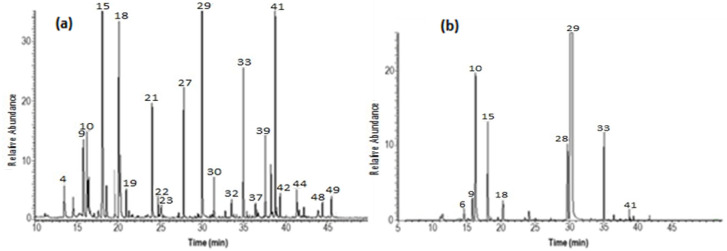
GC-MS TIC chromatogram of (**a**) wild essential oil (WEO_OR) and (**b**) commercial essential oil (CEO_OR) of the plant species *Origanum vulgare*. The numbers shown on each peak correspond to those in Table 1, allowing for the identification of the compounds associated with each peak.

**Figure 2 plants-14-02468-f002:**
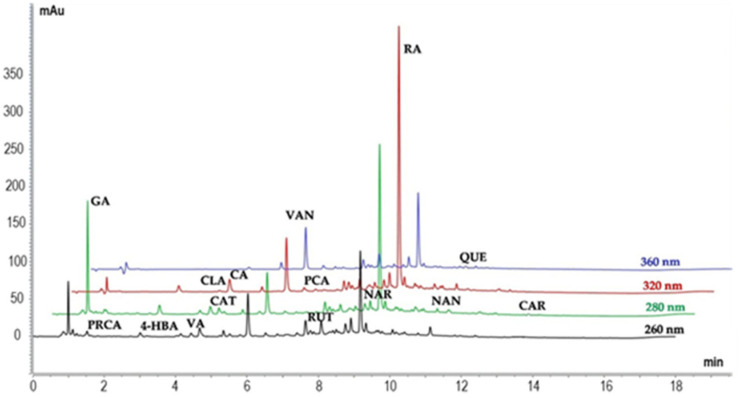
HPLC-DAD profiles at 260, 280, 320, and 360 nm of *O. vulgare* extract.

**Figure 3 plants-14-02468-f003:**
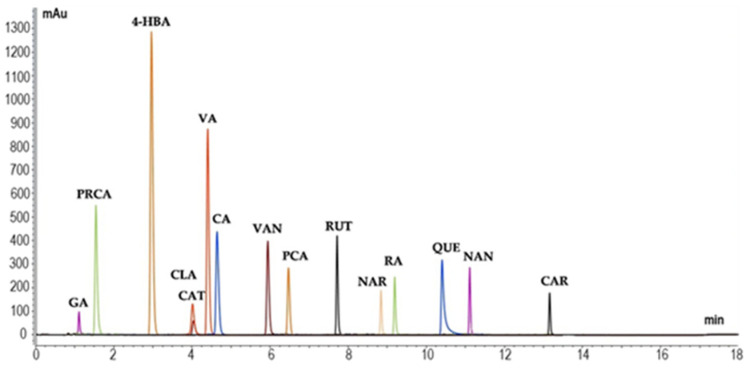
Overlapping HPLC-DAD chromatograms of 15 phenolic standards.

**Figure 4 plants-14-02468-f004:**
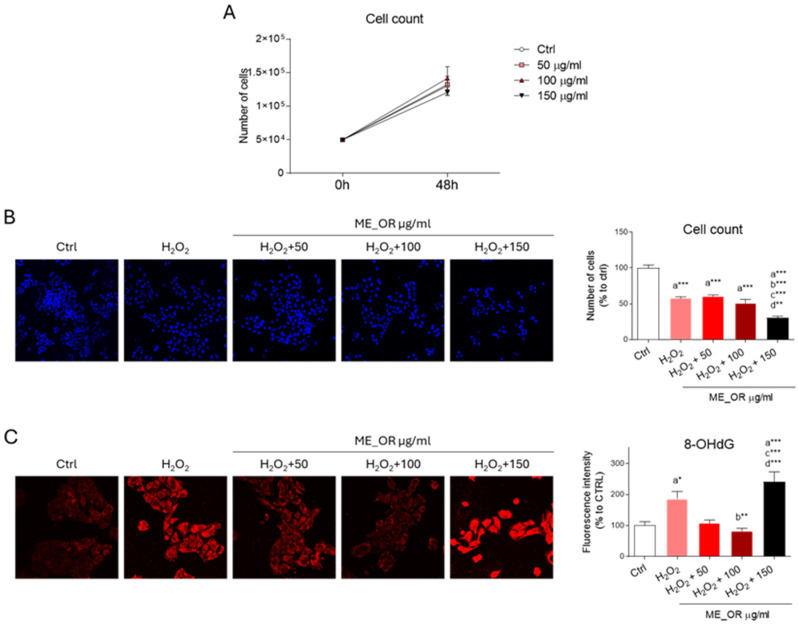
Effects of ME_OR on HepG2 cell viability and oxidative stress. (**A**) Cell proliferation was assessed in HepG2 cells treated with vehicle (DMSO, Ctrl) and ME_OR (50, 100, 150 µg mL^−1^). Data represent the mean ± standard deviation. Statistical analysis was assessed using two-way ANOVA, followed by a Bonferroni post hoc test. (**B**) Representative images and quantification of DAPI-stained nuclei in HepG2 cells treated with vehicle (DMSO, Ctrl), H_2_O_2_ (500 µM), or ME_OR (50, 100, 150 µg mL^−1^) prior to H_2_O_2_-induced oxidative stress. (**C**) Immunofluorescence and confocal analysis for 8-OHdG performed in HepG2 cells stimulated as in (**B**). *n* = 3 different experiments. Statistical analysis was assessed using one-way ANOVA, followed by Tukey’s post hoc test. * *p* < 0.05; ** *p* < 0.01; *** *p* < 0.001. “a” indicates statistical significance versus control group (Ctrl); “b” indicates statistical significance compared to H_2_O_2_ group; “c” indicates statistical significance relative to H_2_O_2_ + ME_OR 50 µg mL^−1^ group; “d” indicates statistical significance against H_2_O_2_ + ME_OR 100 µg mL^−1^ group. Images were acquired using the Leica TCS SP8 confocal microscope and Leica Application Suite X (LAS X) software at 20× (**B**) or 40× (**C**) magnification. Scale bar: 50 µm.

**Figure 5 plants-14-02468-f005:**
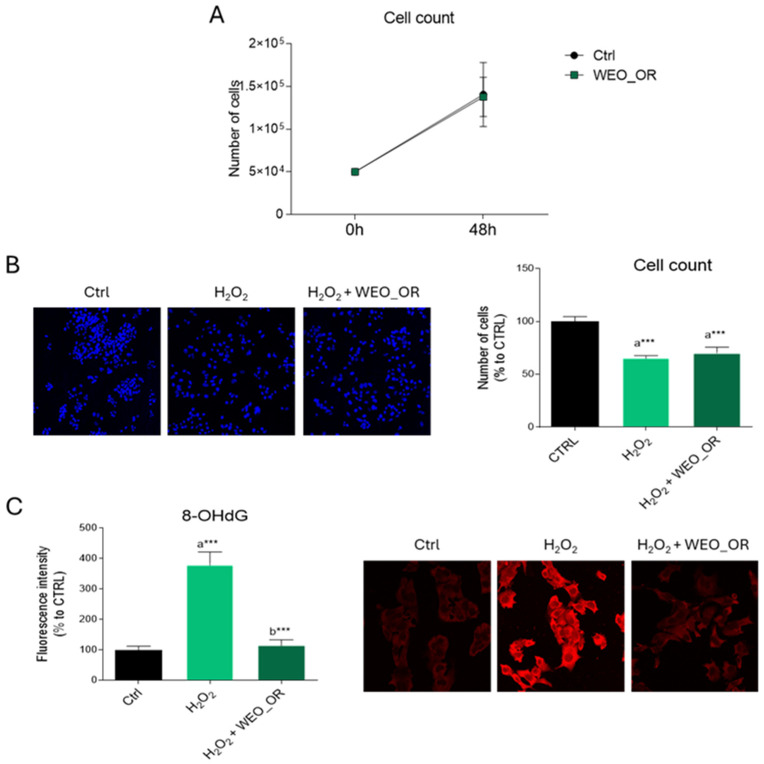
Effects of WEO_OR on HepG2 cell viability and oxidative damage induced by H_2_O_2_. (**A**) HepG2 cell proliferation was measured following treatment with vehicle (DMSO, Ctrl) or WEO_OR (0.005%). Data were analyzed using two-way ANOVA, followed by a Bonferroni post hoc test. (**B**) Visualization and quantification of DAPI-stained nuclei in HepG2 cells treated with vehicle (DMSO, Ctrl), H_2_O_2_ (500 µM), or ME_OR (50, 100, or 150 µg mL^−1^) before exposure to H_2_O_2_. (**C**) Immunofluorescence analysis of 8-OHdG in HepG2 cells treated as in (**B**). *n* = 3 different experiments. Data represent the means ± standard deviation. Results were analyzed by using one-way ANOVA, followed by Tukey’s post hoc test. *** *p* < 0.001. “a” indicates statistical significance compared to control group (Ctrl); “b” indicates statistical significance versus H_2_O_2_ group. Images were acquired using the Leica TCS SP8 confocal microscope and Leica Application Suite X (LAS X) software at 20× (**B**) or 40× (**C**) magnification. Scale bar: 50 µm.

**Table 1 plants-14-02468-t001:** Chemical composition of the wild essential oil by leaves and inflorescences (WEO_OR) and commercial essential oil (CEO_OR) of the plant species *Origanum vulgare*.

N	Compound	Exp RI	Ref RI	Area % ± SD WEO_OR	Area % ± SD CEO_OR	Abbr.
1	*α*-Thujene	929	930	0.23 ± 0.05	0.32 ± 0.03	MM
2	*α*-Pinene	935	939	0.16 ± 0.02	0.51 ± 0.01	BM
3	Camphene	948	954	-	0.06 ± 0.01	BM
4	*β*-Pinene	975	979	1.94 ± 0.14	0.09 ± 0.01	BM
5	Decene	984	989	-	0.09 ± 0.01	OT
6	Myrcene	993	990	0.99 ± 0.02	0.58 ± 0.03	AM
7	*α*-Phellandrene	1002	1002	-	0.11 ± 0.02	MM
8	3-Carene	1008	1011	-	0.05 ± 0.01	BM
9	*α*-Terpinene	1016	1017	3.72 ± 0.13	1.18 ± 0.03	MM
10	*p*-Cymene	1024	1024	3.47 ± 0.13	6.81 ± 0.17	MM
11	*o*-Cymene	1030	1026	-	0.35 ± 0.03	MM
12	Limonene	1030	1029	1.49 ± 0.18	-	MM
13	(*Z*)-*b*-Ocimene	1043	1037	0.19 ± 0.02	-	AM
14	(*E*)-*b*-Ocimene	1053	1050	0.32 ± 0.11	-	AM
15	γ-Terpinene	1061	1059	17.25 ± 0.60	3.70 ± 0.16	MM
16	*p*-Mentha-3.8-diene	1070	1072	1.26 ± 0.10	0.12 ± 0.01	MM
17	Terpinolene	1089	1088	1.91 ± 0.04	0.14 ± 0.01	MM
18	Linalool	1100	1096	9.7 ± 0.36	0.78 ± 0.04	AMO
19	Octen-3-yl-acetate	1116	1112	0.89 ± 0.08	-	OT
20	Borneol	1168	1169	-	0.18 ± 0.00	BMO
21	Terpinen-4-ol	1179	1177	3.97 ± 0.05	0.58 ± 0.02	MMO
22	*α*-Terpineol	1192	1188	0.65 ± 0.03	0.02 ± 0.02	MMO
23	Methyl cavicol (Estragole)	1198	1196	0.47 ± 0.28	-	OT
24	γ-Terpineol	1197	1199	-	0.09 ± 0.00	MMO
25	Dihydrocarvone	1205	1200	0.10 ± 0.03	0.04 ± 0.01	MMO
26	Carvacrol-methyl ether	1247	1244	0.19 ± 0.04	0.09 ± 0.01	MMO
27	Linalyl acetate	1260	1257	4.18 ± 0.15	-	AMO
28	Thymol	1298	1290	-	4.39 ± 0.06	MMO
29	Carvacrol	1305	1299	19.17 ± 0.32	75.95 ± 0.40	MMO
30	δ-Elemene	1339	1338	1.19 ± 0.14	-	MS
31	Neryl acetate	1369	1361	0.23 ± 0.02	-	AMO
32	Geranil acetate	1387	1381	0.50 ± 0.08	-	AMO
33	*β*-Caryophyllene	1420	1419	4.79 ± 0.11	2.28 ± 0.03	BS
34	*β*-Copaene	1431	1432	0.21 ± 0.02	-	BS
35	*trans α*-Bergamotene	1438	1434	0.08 ± 0.01	-	BS
36	Aromadendrene	1446	1441	0.11 ± 0.01	-	BS
37	*allo* Aromadendrene	1455	1460	0.48 ± 0.02	0.14 ± 0.02	BS
38	*trans β*-Farnesene	1460	1456	0.21 ± 0.05	-	AS
39	Germacrene D	1483	1485	2.88 ± 0.15	-	MS
40	*γ*-Amorphene	1498	1495	2.02 ± 0.20	-	BS
41	Germacrene A	1510	1509	7.11 ± 0.19	0.29 ± 0.02	MS
42	*δ*-Amorphene	1516	1512	0.32 ± 0.02	-	BS
43	*δ*-Cadinene	1526	1523	0.77 ± 0.07	0.14 ± 0.01	BS
44	Spatulenol	1581	1578	0.51 ± 0.04	-	BSO
45	Caryophyllene-oxide	1586	1583	0.30 ± 0.03	0.19 ± 0.02	BSO
46	Diethyl phthalate	1599	1590	0.29 ± 0.14	-	OT
47	*α*-Muurolol	1645	1646	0.45 ± 0.01	-	BSO
48	Eudesmol 7-epi-*α*	1659	1663	0.60 ± 0.06	-	BSO
49	Cedren-13-ol 8	1687	1689	0.85 ± 0.08	-	BSO

Abbreviations: AM: aliphatic monoterpenes; MM: monocyclic monoterpenes; BM: bi- and tricyclic monoterpenes; AMO: aliphatic monoterpenoids; MMO: monocyclic monoterpenoids; BMO: bi- and tricyclic monoterpenoids; AS: aliphatic sesquiterpenes; MS: monocyclic sesquiterpenes; BS: bi- and tricyclic sesquiterpenes; ASO: aliphatic sesquiterpenoids; MSO: monocyclic sesquiterpenoids; BSO: bi- and tricyclic sesquiterpenoids; OT: others; SD: standard deviation; Exp. RI: experimental retention index; Ref. RI: literature data.

**Table 2 plants-14-02468-t002:** List of terpenes in the wild essential oil (WEO_OR) and commercial essential oil (CEO_OR) of the plant species *Origanum vulgare*.

	Abbreviation	Area % WEO_OR	Area % CEO_OR
Aliphatic monoterpenes	AM	1.5	0.58
Monocyclic monoterpenes	MM	29.33	12.73
Bi- and tricyclic monoterpenes	BM	2.1	0.71
**Monoterpenes**	**M**	**32.93**	**14.02**
Aliphatic oxygenated monoterpenes	AMO	14.61	0.78
Monocyclic oxygenated monoterpenes	MMO	24.08	81.16
Bi- and tricyclic oxygenated monoterpenes	BMO	-	-
**Oxygenate monoterpenes**	**MO**	**38.69**	**81.94**
Aliphatic sesquiterpenes	AS	0.21	-
Monocyclic sesquiterpenes	MS	11.18	0.29
Bi- and tricyclic sesquiterpenes	BS	8.78	2.56
**Sesquiterpenes**	**S**	**20.17**	**2.85**
Aliphatic oxygenated sesquiterpenes	ASO	-	-
Monocyclic oxygenated sesquiterpenes	MSO	-	-
Bi- and tricyclic oxygenated sesquiterpenes	BSO	2.71	0.19
**Oxygenate sesquiterpenes**	**SO**	**2.71**	**0.19**
**Others**	**OT**	**1.65**	**0.00**

Abbreviations: AM: aliphatic monoterpenes; MM: monocyclic monoterpenes; BM: bi- and tricyclic monoterpenes; AMO: aliphatic oxygenated monoterpenes; MMO: monocyclic oxygenated monoterpenes; BMO: bi- and tricyclic oxygenated monoterpenes; AS: aliphatic sesquiterpenes; MS: monocyclic sesquiterpenes; BS: bi- and tricyclic sesquiterpenes; ASO: aliphatic oxygenated sesquiterpenes; MSO: monocyclic oxygenated sesquiterpenes; BSO: bi- and tricyclic oxygenates sesquiterpenes; OT: others.

**Table 3 plants-14-02468-t003:** Content of phenolic compounds in ME_OR.

Peak	Phenolic Compounds	Abbr.	RT	λ (nm)	ME_OR ± SD (mg g^−1^ DW)	R^2^
1	Gallic acid	GA	1.1	280	13.1 ± 1.3	0.9998
2	Protocatechuic acid	PRCA	1.5	260	1.2 ± 0.4	0.9989
3	4-Hydroxibenzoic acid	4-HBA	3.0	260	0.4 ± 0.1	0.9997
4	Chlorogenic acid	CLA	4.0	320	0.2 ± 0.1	0.9996
5	Catechin	CAT	4.1	280	2.2 ± 0.2	0.9990
6	Vanillic acid	VA	4.4	260	0.9 ± 0.2	0.9991
7	Caffeic acid	CA	4.6	320	1.1 ± 0.1	0.9988
8	Vanillin	VAN	6.0	280	8.1 ± 0.6	0.9989
9	*p*-Coumaric acid	PCA	6.5	320	0.4 ± 0.1	0.9998
10	Rutin	RUT	7.7	260	5.2 ± 0.5	0.9993
11	Naringin	NAR	8.8	280	4.4 ± 0.3	0.9996
12	Rosmarinic acid	RA	9.2	320	38.8 ± 2.8	0.9983
13	Quercetin	QUE	10.4	260	1.7 ± 0.4	0.9986
14	Naringenin	NAN	11.2	280	1.2 ± 0.3	0.9987
15	Carvacrol	CAR	13.2	280	0.3 ± 0.1	0.9997

RT: retention time; R^2^: coefficient of determination.

**Table 4 plants-14-02468-t004:** Antioxidant activity of wild essential oil (WEO_OR), commercial essential oil (CEO_OR), and methanolic extract of Origanum vulgare (DPPH, ABTS, and FRAP assays).

Sample	DPPH	ABTS		FRAP
	IC_50_ (mg mL^−1^)	IC_50_ (mg mL^−1^)	(mg TE mL^−1^)	(mg TE g^−1^)
**ME_OR**	0.052 ± 0.01	0.044 ± 0.006	3.94 ± 0.07	30.58 ± 3.7
**CEO_OR**	0.45 ± 0,11	0.033 ± 0.015	9.57 ± 0.52	7.33 ± 0.31
**WEO_OR**	1.54 ± 0.22	0.56 ± 0.07	0.10 ± 0.02	-
**Positive control**	0.00375 *	0.00347 **		0.00506 **

IC_50_: concentration of the extract that inhibits 50% of the radical activity. TE: Trolox equivalents. Each value is a mean ± SD of triplicate analysis. * Ascorbic acid; ** Trolox equivalent.

**Table 5 plants-14-02468-t005:** Total phenolic and flavonoid content of methanolic extract of *Origanum vulgare* (ME_OR).

Sample	Total Polyphenols	Flavonoids	
	(mg g^−1^ GAE)	(mg QUE g^−1^ DM)	(mg CAE g^−1^ DM)
**ME_OR**	75.49 ± 0.9	147.2 ± 2.1	34.7 ± 0.5

GAE: gallic acid equivalents; QUE: quercetin equivalents; CAE: catechin equivalents. Each value is a mean ± SD of a triplicate analysis.

**Table 6 plants-14-02468-t006:** Gradient program applied for the UHPLC analysis.

Time	Flow (mL/min)	% B
0	0.4	5
5	0.4	16
8	0.4	30
14	0.4	85
16	0.4	5
20	0.4	5

## Data Availability

The original contributions presented in the study are included in the article; further inquiries can be directed to the corresponding author.

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
