# Peer review of "Phytochemistry and Bioactivity of Essential Oil and Methanolic Extracts of Origanum vulgare L. from Central Italy"

_plants, 2025, doi:10.3390/plants14162468_

Round 1

Reviewer 1 Report

Comments and Suggestions for Authors

Dear authors,

I have made some suggestions with comments directly in the paper.

Author Response

Reviewer 1#

Thank you very much for your suggestions. We reported in the revised manuscript your requests in red

Comments and Suggestions for Authors:

In your results and in the introduction, the objective appears to be the comparison between oregano wild essential oil and commercial essential oil; however, this is not clearly stated here. Please add this information.

Reply by the authors: Thank you for this valuable suggestion. We agree that the comparative aspect between the wild and commercial essential oils is a key point of the study. We have therefore revised the abstract to make this objective explicit.

Please add References to support these statements

Reply by the authors: As requested, the bibliographic reference has been added [30]: Mugao, L. Factors influencing yield, chemical composition and efficacy of essential oils. International Journal of Multidisciplinary Research and Growth Evaluation. 2024, 05, 169-178. doi.org/10.54660/.IJMRGE.2024.5.4.169-178.

And regarding the acquisition of commercial essential oil mentioned earlier?

Reply by the authors: Thank you for your observation. As correctly noted, the commercial essential oil (WEO_COM) was not obtained through in-house extraction. It was purchased from a certified supplier and analyzed in its original form after appropriate dilution, without further processing. This information has now been clarified in the Materials and Methods, 4.3 section.

The methodology for antioxidant activity using ABTS suggests that spectrophotometric measurements should be taken between 1 and 6 minutes after the reaction between the sample and ABTS (RE et al., 1999). What is the reference used that indicates a 30-minute wait? What is the justification for this?

Reply by the authors: We thank the Reviewer for the valuable comment. We acknowledge that the manuscript incorrectly stated an incubation time of 30 minutes for the ABTS assay. We confirm that the actual incubation time used in our experimental procedure was 5 minutes at room temperature.

This choice is consistent with the original method described by Re et al. (1999), which recommends taking absorbance readings between 1 and 6 minutes, depending on the antioxidant activity of the sample. In our case, the oregano extract exhibited a rapid reaction kinetics with the ABTS•+ radical, similar to what is reported for other plant-derived antioxidant extracts. We selected a 5-minute incubation time as a widely accepted compromise between reaction completeness and operational efficiency. Importantly, we observed that the absorbance value had stabilized within this time frame, indicating that the reaction had reached a plateau and the measurement was reliable.

We have corrected the text in the revised manuscript to accurately reflect the procedure adopted.

Reviewer 2 Report

Comments and Suggestions for Authors

A well-designed study with comprehensive biological evaluations and rigorous analytical techniques is presented in the manuscript "Phytochemistry and Bioactivity of Essential Oil and Methanolic Extracts of Origanum vulgare L. from Central Italy." GC–MS for essential oil profiling, UHPLC-DAD for phenolic quantification, and in vitro bioactivity assessments provide a comprehensive multidisciplinary approach. A structured and critical assessment of the manuscript is provided below.

  1. Abstract:
    - Provide additional quantitative data, such as the percentage of antioxidant reduction and the ICâ‚…â‚€ values.
    - Statistical significance thresholds should be included.
  2. Introduction
    - Explain the rationale behind the selection of HepG2 cells over other oxidative stress models.
    - Provide a more detailed explanation of the novelty: example, is this the first comparative profiling of natural and commercial EO from central Italy?
  3. Antioxidant assays
    • - WEO_OR lacks FRAP data. If the item is unavailable, please provide an explanation
    • Statistical analysis must be performed to state the difference bioactivity of each fraction.
    • The solubility of each fraction influence the bioactivity tested by DPPH, ABTS and FRAP or not? This must be considered when non-polar extract is used.
  4. Figures 4 & 5: Ensure microscopy images are high resolution and properly labeled (arrows, scale bars, magnification).

  5. Figure4: If it is possible that the extract at 150 ug/mL synergistic with H2O2 to increase ROS and cell death? Please discuss.
  6. Discussion 

    1. Incorporate additional mechanistic information regarding the antioxidant function of rosmarinic acid at the molecular or gene expression level, such as the activation of Nrf2 and the modulation of metabolic pathways.
    2. Discuss the implications of chemotype classification for commercial standardization practices.
    3. It is crucial to consider the phenomenon of pro-oxidant activity at high phenolic concentrations. I suggest that redox imbalance be considered as a potential contributor through Fenton-like mechanisms and reactive quinones.

Author Response

Reviewer 2#

A well-designed study with comprehensive biological evaluations and rigorous analytical techniques is presented in the manuscript "Phytochemistry and Bioactivity of Essential Oil and Methanolic Extracts of Origanum vulgare L. from Central Italy." GC–MS for essential oil profiling, UHPLC-DAD for phenolic quantification, and in vitro bioactivity assessments provide a comprehensive multidisciplinary approach. A structured and critical assessment of the manuscript is provided below.

We thank the reviewer for his positive comments and for the time he devoted to reviewing the manuscript. The corrections to the revised manuscript are listed below and reported in the new version (in red).

Abstract:

- Provide additional quantitative data, such as the percentage of antioxidant reduction and the ICâ‚…â‚€ values.

Replay by the authors: Thank you for the suggestion. We have revised the abstract to include ICâ‚…â‚€ values.

- Statistical significance thresholds should be included.

Replay by the authors: Statistical significance thresholds were reported in Statistical Analysis section 4.11.  and in figure captions 4 and 5.

Introduction

- Explain the rationale behind the selection of HepG2 cells over other oxidative stress models.

Reply by the authors: We thank the reviewer for giving us the opportunity to further clarify the rationale behind the selection of our experimental model. To assess the biological activity of the extracts of interest, we used HepG2 cells, as they are widely employed as a cell culture model to study cytotoxicity and the response to oxidative stress (Zhao et al., 2021; Wu et al., 2018; Gong et al., 2006; PG et al., 2024; Huang et al., 2025; Mao et al., 2024; Pfeifer et al., 2025; Luo et al., 2024). Although this point was already mentioned at the end of the “Introduction” section, to further strengthen the reason of choosing this model, we have now added other supporting references, including a review that explicitly indicates HepG2 cells as a commonly used cell culture model for studying antioxidant capacity (Zhang et al., 2017; Babu et al., 2017). Please see “Introduction” section (“As an experimental model we used HepG2, a robust cell line that ensures experimental reproducibility, as it is commonly employed to study cytotoxicity and response to oxidative stress”).

- Provide a more detailed explanation of the novelty: example, is this the first comparative profiling of natural and commercial EO from central Italy?

Reply by the authors: Thank you for your comment. While several studies have compared wild versus cultivated oregano essential oils such as Ilić et al., 2022, who reported chemical and antioxidant differences between wild and cultivated populations in Serbia (Ilić, Z.; Stanojević, L.; Milenković, L.; Šunić, L.; Milenković, A.; Stanojević, J.; Cvetković, D. The Yield, Chemical Composition, and Antioxidant Activities of Essential Oils from Different Plant Parts of the Wild and Cultivated Oregano (Origanum vulgare L.). Horticulturae 2022, 8, 1042. https://doi.org/10.3390/horticulturae8111042) and others have profiled wild oregano oils from southern Italy (e.g., Campania and Sicily) in terms of composition and antimicrobial activity De Martino, L.; De Feo, V.; Formisano, C.; Mignola, E.; Senatore, F. Chemical Composition and Antimicrobial Activity of the Essential Oils from Three Chemotypes of Origanum vulgare L. ssp. hirtum (Link) Ietswaart Growing Wild in Campania (Southern Italy). Molecules 2009, 14, 2735-2746. https://doi.org/10.3390/molecules14082735 ) no previous work has performed a direct comparison between a wild-harvested central Italian oregano essential oil (WEO_OR) and a commercially available oregano oil (CEO_OR).

Antioxidant assays

- WEO_OR lacks FRAP data. If the item is unavailable, please provide an explanation

Reply by the authors: We thank the Reviewer for the observation regarding the lack of FRAP data for the WEO_OR sample. Unfortunately, due to the limited quantity of the extract available and the extensive number of analyses performed, the material was entirely consumed before FRAP analysis could be reliably completed. While a preliminary measurement was performed, we were unable to replicate it in triplicate, and therefore chose not to report unconfirmed data, in accordance with good scientific practice.

The solubility of each fraction influence the bioactivity tested by DPPH, ABTS and FRAP or not? This must be considered when non-polar extract is used.

Reply by the authors: Thank you for raising this important point. We agree that the solubility and polarity of each extract may influence the measured antioxidant activity, particularly in spectrophotometric assays such as DPPH, ABTS, and FRAP. In our study, both the methanolic extract (ME_OR) and the essential oils (WEO_OR and CEO_OR) were diluted in methanol prior to testing. While the methanolic extract was fully soluble in the reaction environment, we acknowledge that essential oils, being non-polar, may exhibit limited miscibility in aqueous-based systems. This partial immiscibility could contribute to an underestimation of their true antioxidant potential when compared to the polar extract. Although no solubility issues were visually observed during the assays, we recognize this as a possible limitation and appreciate the reviewer’s observation.

Figures 4 & 5: Ensure microscopy images are high resolution and properly labeled (arrows, scale bars, magnification).

Reply by the authors: We apologize to the reviewer for this oversight. We have now ensured that all images are of high resolution. In case there are any issues with resolution loss due to file conversion or upload, we would be happy to provide the original, image files, not embedded in the text. Additionally, we have included scale bars on the confocal images and described them in the figure legends. Furthermore, we added details about the image acquisition in the “Results” section (“Images were captured using Leica TCS SP8 equipped with 20× or 40× magnification and analyzed with Leica LAS X Software (version 3.5.5) for Windows 10”).

Figure4: If it is possible that the extract at 150 ug/mL synergistic with H2O2 to increase ROS and cell death? Please discuss.

Reply by the authors: The reviewer has raised an interesting point regarding the dosage: indeed, the highest dose appears to worsen the oxidative damage induced by H2O2. Although we are currently unable to provide a definitive explanation for this phenomenon, we have suggested a possible interpretation, supported by literature evidence, in the “Discussion” section (“Furthermore, it must be emphasized that the effects observed in this work on methanolic extract are strictly dependent on the dose. Indeed, at high concentrations, molecules known for their antioxidant action lose their effects and manifest a pro-oxidant action, causing membrane and DNA damage. Conversely, at low concentrations, these molecules can manifest a protective effect on the membrane and DNA, thus acting as antioxidants. In this context, the low-dose methanolic extract (50-100 ug mL-1) showed an antioxidant effect, while the higher dose (150 ug mL-1) induced an opposite effect, further in-creasing levels of the oxidative damage marker 8-OHdG, thus acting as a pro-oxidant. Furthermore, the highest concentration tested not only exacerbated the intracellular oxidative stress, but also increased the susceptibility of HepG2 cells to Hâ‚‚Oâ‚‚-induced cytotoxicity. Similar results were also obtained by other research groups that focused their attention on the effects of carvacrol and thymol, two compounds particularly enriched in oregano essential oil. Specifically, an increase in ROS was observed in Caco2 cells after 24 hours of exposure to a concentration of 460 uM carvacrol and after 48 hours of treatment at 230 uM. A similar observation was obtained by the administration of a carvacrol-thymol mixture in a 10:1 ratio. Furthermore, other studies have reported that carvacrol at concentrations above 380 uM induces increased levels of the oxidative damage markers MDA and 8-OHdG in lung cancer cells (H1299) 24 and 48 hours after administration”).

Discussion

Incorporate additional mechanistic information regarding the antioxidant function of rosmarinic acid at the molecular or gene expression level, such as the activation of Nrf2 and the modulation of metabolic pathways.

Reply by the authors: We thank the reviewer for giving us the opportunity to better integrate our findings within the context of the current state of the art. Indeed, including some details regarding the mechanisms linking rosmarinic acid (the most abundant phenolic compound identified in our extract) to the antioxidant activity, helps to contextualize our work more effectively within the framework of the existing knowledge. In the revised version, we have therefore discussed the antioxidant mechanisms of rosmarinic acid, with particular emphasis on the activation of the NRF2-mediated pathway (please see “Discussion” section: “Concerning the most abundant compound, several findings have demonstrated that rosmarinic acid efficiently counteracts oxidative stress in diverse pre-clinical experimental models. Interestingly, it has been reported that rosmarinic acid exerts its antioxidant effects primarily through the activation of the nuclear factor erythroid 2–related factor 2 (NRF2), one of the master transcription factors governing redox homeostasis. Although the upstream mechanisms are not still completely elucidated and may vary on the specific physiopathological context, it has been extensively demonstrated that rosmarinic acid can promote NRF2 stabilization/activation, which migrates to the nucleus to promote the expression of antioxidant response element (ARE)-driven genes such as heme oxygenase 1 (HO-1) and Glutamate-Cysteine Ligase Modifier Subunit (GCLM), thereby enhancing intracellular defense. Accordingly, rosmarinic acid administration restores GSH/GSSG balance and strongly suppresses ROS production. Collectively, this evidence highlights that rosmarinic acid contrasts oxidative stress not only by acting as a direct scavenger, but also by upregulating the endogenous antioxidant systems”).

Discuss the implications of chemotype classification for commercial standardization practices.

Reply by the authors: Thank you for this valuable observation. We agree that chemotype classification plays a crucial role in the standardization and commercial quality control of essential oils. The essential oil from wild Origanum vulgare analyzed in our study was characterized by a carvacrol chemotype, as the dominant compound but also containing relevant amounts of sesquiterpenes and other minor constituents. This chemical profile differs from that of the commercial essential oil, which showed a more simplified composition with a higher proportion of carvacrol, likely resulting from selective cultivation or standardization practices. These differences underscore the importance of chemotype identification in ensuring consistency, reproducibility, and efficacy in commercial products. We have now briefly discussed this point in the revised Discussion section.

Round 2

Reviewer 2 Report

Comments and Suggestions for Authors

The author well respond to the comments and suggestions. This revised version can be now considered to be accepted for the publication.